# Effects of a Cooling Vest on Core and Skin Temperatures Following a Soccer-Specific Exercise Protocol [note 1]

**DOI:** 10.3390/sports13070235

**Published:** 2025-07-17

**Authors:** Carlos Lorente-González, Jose Vicente Beltran-Garrido, Abraham Batalla-Gavaldà, Francisco Corbi

**Affiliations:** 1Institut Nacional d’Educació Física de Catalunya, Facultat de Lleida, Universitat de Lleida, Complex de La Caparrella, s/n, 25192 Lleida, Spain; clorent2@xtec.cat; 2Department of Education Science, School of Humanities and Communication Sciences, Universidad Cardenal Herrera-CEU, CEU Universities, 12006 Castellón de la Plana, Spain; 3Physical Exercise and Performance Research Group, Universidad Cardenal Herrera-CEU, CEU Universities, 12006 Castellón de la Plana, Spain; 4University School of Health and Sport (EUSES), Universitat Rovira i Virgili, 43870 Amposta, Spain; a.batalla@euseste.es; 5Department of Education and Specific Didactics, Faculty of Humanities and Social Sciences, Universitat Jaume I, 12071 Castellón de la Plana, Spain; 6Department of Clinical Sciences, Faculty of Medicine and Health Sciences, University of Barcelona, 08907 L’Hospitalet de Llobregat, Spain; fcorbi@ub.edu

**Keywords:** cool vest, cryotherapy, soccer, recovery, skin temperature

## Abstract

Background/Objectives: Cooling strategies are critical for optimizing athlete recovery and performance in team sports, yet practical, accessible solutions remain underexplored. This study evaluated the efficacy of a cool vest during a 15 min recovery period following 45 min of simulated soccer match play on core and skin temperature regulation. Methods: Forty-seven physically active males completed an exercise protocol replicating the physiological demands of a soccer half-match. Participants were randomized into an experimental group using a cool vest during recovery (IG, n = 24) or a control group (CG, n = 23) without cooling intervention. Results: Compared to CG, IG exhibited small but significant reductions in skin temperature (31.46 ± 0.67 °C vs. 32.40 ± 1.04 °C; ES = −0.24, 95% CI: −0.40 to −0.08; *p* = 0.003) and tympanic temperature (35.99 ± 0.45 °C vs. 36.54 ± 0.66 °C; ES = −0.43, 95% CI: −0.67 to −0.18; *p* < 0.001) 10 min post-exercise. These differences intensified to small and moderate effects at 15 min post-exercise for skin (31.07 ± 0.67 °C vs. 32.51 ± 0.91 °C; ES = −0.37, 95% CI: −0.53 to −0.21; *p* < 0.001) and tympanic temperatures (35.70 ± 0.42 °C vs. 36.65 ± 0.58 °C; ES = −0.71, 95% CI: −0.96 to −0.46; *p* < 0.001), respectively, with IG maintaining consistently lower values. No temperature changes were observed in CG. Conclusions: These findings demonstrate that a 15 min cool vest application during halftime effectively reduces core and skin temperatures, compared to passive recovery. This supports its utility as a practical, low-cost strategy for thermoregulatory management in soccer, warranting further investigation into its long-term performance benefits.

## 1. Introduction

In sports performance, cooling effects have been studied to lower core body temperature, enhancing heat storage capacity and improving performance [1]. Some findings suggest that thermoregulatory strain is decreased due to a redistribution of blood flow towards active skeletal muscles, decreasing heart rate and sweat rate, and altering skin and core body temperature, thus increasing thermal comfort [2,3]. These benefits make cooling a common and inexpensive method to enhance exercise performance through different forms of application such as whole-body cryotherapy [4,5], cold-water immersion baths [6], cool vests [7], cold towels [8] and ice slurry ingestion [9].

Moreover, precooling methods have been applied to counteract the negative effects of heat stress on fatigue, enhancing the body’s capacity to store metabolic and environmental heat, and improving submaximal exercise performance [10,11]. Although most studies were conducted under ambient temperatures of 30 °C or higher [12], these findings suggest that cooling may be effective at high environmental temperatures and could also improve thermal comfort while exercising at lower temperatures [13].

Cryotherapy, as a post-exercise therapy, might decrease inflammation in the musculoskeletal system [14], which may potentially reduce symptoms of delayed onset muscle soreness (DOMS) after strenuous exercise [15]. The mechanism is primarily related to its vasoconstrictive effect, reducing the inflammation responses by decreasing cell metabolism [16]. Consequently, cryotherapy could be an interesting strategy during recovery to provide ergogenic benefits, treating muscle syndromes of overuse and reducing recovery time between training sessions resulting from the attenuation of thermal load [17].

For these reasons, both pre- and post-exercise cooling interventions may be considered as a method to enhance performance by improving thermoregulatory and cardiovascular responses—such as heart rate recovery, blood pressure regulation and venous return—through physiological benefits during athletic events, especially during breaks, such as halftimes or between bouts of exercise [18]. Although in recent years several studies have examined the effects of cooling in team sport athletes—such as football [19,20], basketball [21], and rugby [5]—with some reporting positive outcomes in counteracting performance losses in speed and shooting accuracy, as well as reductions in creatine kinase concentrations in both football and rugby, there is still limited knowledge regarding the effects of using a cooling vest during short recovery periods, such as halftime in football. However, sports events are not always conducted under high ambient temperatures, which questions the effect of cooling under low exogenous heat stress conditions.

Although most studies have focused on conditions of high environmental heat stress, it is important to highlight that in prolonged intermittent efforts such as soccer, the body accumulates considerable thermal load even in temperate environments, due to sustained elevations in core temperature and limited opportunities for heat dissipation during intense activity. This thermoregulatory strain is a key modulator of both central and peripheral fatigue [22], contributing to impairments in physical and technical performance [23], particularly during the second half of play. Research has shown that when core body temperature exceeds 39 °C, neuromuscular function declines [24], technical skill execution becomes less accurate, and decision-making may be compromised [25]. In addition, prolonged exposure to elevated thermal stress without effective countermeasures may also pose potential health risks [26]. Therefore, cooling strategies applied during recovery periods, such as halftime, may help attenuate thermal strain, support recovery, and sustain performance [27], even under moderate ambient conditions.

On this basis, we hypothesized that a cool vest applied during halftime (15 min) of a soccer match could provide physiological benefits, enhancing recovery and attenuating fatigue-induced thermal strain in a temperate environment. Therefore, the aim of this study was to determine the effect of a cool vest during the 15 min break after a soccer-specific protocol on core and skin temperatures.

## 2. Materials and Methods

### 2.1. Experimental Approach to the Problem

To determine the effect of wearing a cool vest during the 15 min recovery period after a 45 min soccer-specific protocol on core and skin temperature, the whole sample performed an exercise protocol on a treadmill designed to simulate the demands of a soccer match. Then, during a 15 min recovery period, participants were allocated to two groups stratified according to maximal oxygen uptake (VO_2_max) values and depending on whether or not they were wearing the cooling vest. During a 15 min recovery period, the intervention group (IG, n = 24) remained seated and wore the cool vest, while the control group (CG, n = 23) remained seated at passive rest without wearing the cool vest. Skin temperature (Ts) was measured following the ISO 9986 protocol at the beginning and end of the warm-up, every 5 min during the match simulation protocol and every 5 min during the 15 min recovery period. Tympanic temperature (Tt), due to the characteristics of the exercise protocol, was only measured at baseline, immediately following the warm-up and prior to the start of the exercise protocol, and every 5 min during the 15 min recovery period.

### 2.2. Participants

The study sample consisted of 47 physically active males (age: 22.3 ± 1.9 years, height: 178.1 ± 4.8 cm, weight: 72.9 ± 4.7 kg, body mass index (BMI): 23.13 ± 1.5 kg·m^−2^, VO_2_max: 56.1 ± 3.6 mL·kg^−1^·min^−1^). The training background of the sample consisted of strength-trained individuals (n = 14), runners (n = 8), handball players (n = 7), cyclists (n = 6), swimmers (n = 5), football players (n = 4) and basketball players (n = 3). The baseline characteristics of both groups are shown in Table 1. The inclusion criteria were as follows: (i) participants did not engage in moderate- or high-intensity physical activity in the 48 h preceding the study, (ii) had no personal or family history of cardiac injury, and (iii) had not suffered any injury that could alter regular sports practice in the last six months prior to the study. None of the participants received any financial or in-kind reward for their collaboration in the study. All participants signed an informed consent form, and a protocol was established for the delivery and explanation of the results. At the time of the study, none of the participants were taking any type of medication, nor were they following a specific dietary pattern or suffering from any respiratory or metabolic disorder. This study was conducted taking into account the principles of the Declaration of Helsinki for human research [28], as well as the criteria established in the Biomedical Research Act [29], and in accordance with the Data Protection Act [30]).

### 2.3. Procedures

#### 2.3.1. Training Interventions

*Intervention group:* The participants in the IG group remained seated at passive rest during the whole 15 min recovery period while wearing a cool vest. The cooling vest (Figure 1) (FlexiFreeze Ice Vest; Maranda Enterprises, LLC. Mequon, WI, USA) employs recooling ice sheets designed to “harness 96 pure water ice cubes into 3 ½ pounds of efficient cooling capability”, per the manufacturer’s recommendations. The vest is light, flexible and is made from neoprene less than 2 cm thick. These vests are equipped with two re-freezable ice sheets, each containing 48 ice cubes in the front and 48 in the back, attached with Velcro for easy replacement. The ice sheets were stored in a freezer for 24 h before use.

*Control group:* Participants of the CG group remained seated at passive rest during the recovery period without wearing a cool vest.

#### 2.3.2. Testing Procedures

*Assessment of maximal oxygen uptake *(VO_2_max): two weeks before the start of the study, an incremental running test [31] was performed on a treadmill by a specialist in sports medicine. The test began with a 2 min warm-up at 6 km·h^−1^, followed by progressive increases of 1 km·h^−1^ every minute, to estimate VO_2_max. The test was conducted in a laboratory room with a constant temperature of 22 ± 1.9 °C and a relative humidity of 30.8 ± 4.8%. To obtain VO_2_max a Quark CPET gas analyzer (Cosmed, Rome, Italy) was used. The criteria for ending the test were the voluntary termination by the subject or exhaustion, defined as the inability to maintain the required pace. Once VO_2_max was determined, each participant was randomly assigned to either experimental or control group, repeating the process until no significant differences were detected between groups.

*Anthropometry*. Body height was evaluated with a Seca Tallimeter^®^ 220 (Seca, Hamburg, Germany, with an accuracy of 0.01 cm) and body weight with a Seca Scale^®^ 700 (Seca, Hamburg, Germany, with an accuracy of 0.05 kg).

*Exercise Protocol*: Three days before the study, participants were instructed not to consume caffeine or alcohol 24 h before each trial, and to avoid eating for 2 h before the test. However, they were encouraged to drink water or other non-caffeinated, non-alcoholic beverages freely before the trials “*ad libitum*”. The exercise protocol consisted of a 45 min intermittent running session on a motorized treadmill, reflecting the time spent at various speed levels based on data from match analysis of Spanish soccer players [32]. The protocol included five 9 min running bouts at speeds of 6.5, 12.5, 16.5, 21, and 23 km·h^−1^, representing 63%, 14%, 15%, 5%, and 3% of the protocol duration, respectively. Intensity levels were organized in a non-cyclical manner in an attempt to reflect the nature of a 45 min soccer game (Figure 2). The protocol was specifically developed for the present study based on the data presented in Di Salvo et al. [32]. In this study, the authors analyzed movement kinematics in professional soccer players to quantify running performance parameters (running distances and speeds) across different playing positions. Similar protocols have been applied recently in other studies, such as Brown et al. [33]. Unfortunately, the use of a motorized treadmill precluded the inclusion of true sprinting, backwards running and sideways activities. Once finished, the participants were allocated to two groups, involving either no cooling or the application of a cool vest during the 15 min seated recovery (simulating halftime). The exercise protocol was conducted in a laboratory room under a constant temperature of 22 ± 1.9 °C and 30.8 ± 4.8% relative humidity.

*Skin Temperature (Ts)*: The skin temperature was registered using an Infrared Thermometer Iberia PCE—894 (PCE Ibérica, Tobarra, Albacete, Spain) with a resolution of 0.1 °C (measuring range −50 °C to +1850 °C). The precision of measurement in the temperature range of our test was ±0.5% of ±2 °C range. The protocol was designed according to ISO 9986 [34]. The skin temperature was recorded from the right scapula, left chest, upper left arm, right anterior thigh and left calf (Figure 1). Specifically, Ts was calculated based on Ramathan’s equation: 0.3 (Chest + Arm) + 0.2 (Thigh + Calf) [35]. Data collection was carried out before the warm-up (baseline), after warm-up (minute 0 of 45 min simulated soccer match), every 5 min during the exercise protocol (5, 10, 15, 20, 25, 30, 35, 40, 45) and in minutes 5, 10 and 15 of the recovery period.

*Tympanic Temperature (Tt)*: The tympanic temperature was measured using an Infrared Thermometer Ri-Thermo^®^ N. (Bruckstraße, Jungingen, Germany). The thermometer detected the infrared radiation generated by the eardrum and the surrounding tissue. The technical error of measurements amounted to 0.2 for temperatures in the range of 32.0 °C to 42.2 °C. The data record was collected before the warm-up (baseline), after warm-up (minute 0 of 45 min simulated soccer match), at the end of the protocol (45 min) and in minutes 5, 10 and 15 of the recovery period.

### 2.4. Sample Size and Power Analysis

The sample size was initially calculated to detect an effect size of d = 2.65, calculated using data from Chaen et al. [36] (IG: 33.76 ± 1.08 °C, n = 8; CG: 36.14 ± 0.67 °C, n = 8), with a power of 0.80 and a significance level of 0.05, which required 5 participants per group. Ultimately, the sample size was considerably higher, with 24 participants in the IG and 23 participants in the CG. To assess the impact of this increase on the sample size, an inverse power analysis was performed using the powerlmm package in R. Study parameters included a significance level (α) of 0.05, a power (1 − β) of 0.80, 14 repeated measurements per subject, and an intraclass between-subject variance of 0.5. The analysis determined that with the available sample sizes, the study has sufficient statistical power to detect a minimum effect size of d = 0.604.

### 2.5. Statistical Analyses

To confirm the data normality of each dataset, the Kolmogorov–Smirnov test, the Q-Q plot of residuals and the random coefficient histogram were used.

To assess the between-group differences of skin and tympanic temperatures during the different time points of the soccer-specific protocol and the recovery period, mixed model analyses were used. The model used for each dependent parameter (i.e., Skin temperature and Tympanic temperature) took group (i.e., IG or CG) and time of the exercise protocol and recovery period (i.e., Baseline, Warm-up, 5 min, 10 min, 15 min, 20 min, 25 min, 30 min, 35 min, 40 min, 45 min, Post 5 min, Post 10 min and Post 15 min) as independent fixed factors and random intercepts on the individual participant. A log-likelihood ratio test was used to assess the goodness of fit of the models. To assess the effects of the cool vest on the dependent parameters at different time points, simple effects were calculated with Group as the simple effect variable and Time as the moderator variable. Standardized mean difference Cohen’s *d* effect sizes were obtained and were interpreted as follows: <0.2 = trivial; 0.2–0.6 = small; 0.6–1.2 = moderate; 1.2–2.0 = large; >2.0 = very large [37]. Statistical significance was established at α < 0.05. Unless otherwise stated, all values are presented as mean ± SD. Data analysis was performed using JAMOVI for Mac (version 2.6.44; The Jamovi project [38]) and the GAMLj3 jamovi module: General analyses for linear models (version 3.6.0) [39].

## 3. Results

The summary of changes of temperature for both groups at different time points are shown in Table 2. The effect sizes of the between-group differences during the recovery time are shown in Figure 2.

Changes of temperature at different time points showed a main effect in both skin (*F* = 47.38, *p* < 0.001) and tympanic temperature (*F* = 11.04, *p* < 0.001). Furthermore, significant interactions were reported in both skin (*F* = 5.44, *p* < 0.001) and tympanic temperature (*F* = 4.43, *p* < 0.001). See Table 2.

Regarding between-group differences of skin temperature, simple effects revealed non-significant differences (*p* > 0.05) at all time points except at Post 10 min (*p* = 0.003; ES = −0.24 (−0.40, −0.08, small)) and Post 15 min (*p* < 0.001; ES = −0.37 (−0.53, −0.21, small)), when the temperature was lower in the IG group. See Figure 3.

Regarding between-group differences of tympanic temperature, simple effects revealed non-significant differences (*p* > 0.05) at all time points except at Post 10 min (*p* < 0.001; ES = −0.43 (−0.67, −0.18, small)) and Post 15 min (*p* < 0.001; ES = −0.71 (−0.96, −0.46, moderate)), when the temperature was lower in the IG group. See Figure 3.

## 4. Discussion

The main finding of this study was the existence of significant differences in both skin and tympanic temperature at 10 and 15 min into the recovery period when the cool vest was used. These results align with other studies that found the same outcome [40,41], which might be limited in environments of high ambient humidity, reducing the body’s ability to dissipate heat [42].

While increased catecholamine as noradrenaline released during high-intensity exercise contributes to systemic vasoconstriction in non-active body areas, which helps maintain blood pressure and redirect blood flow to active muscles, at the same time it reduces skin blood flow, which impairs the ability to dissipate heat through the skin [43]. Although in general, skin temperature seems to remain lower during exercise than in baseline or warm-up, we observed a tendency to decrease as the exercise progresses; some participants seemed to have an individual response depending on the predominance of vasoconstriction or vasodilation or vice versa [44]. This fact indicates the existence of individual response patterns. Therefore, different people exposed to the same temperature may display different stress responses. This could be attributed to the individual thermoregulatory adaptation, but also to other different factors such as aging, gender, type of exercise, intensity, duration, and level of physical fitness that have a substantial effect on this response due, in part, to muscle mass and subcutaneous fat layer [45], which depends on physical fitness and affects thermoregulation [46] through the evaporation of sweat and circulatory response [47]. These differences in body size, configuration and composition may partly explain individual responses associated with cold exposure. For these reasons, it seems logical to think that it is necessary to assess their effects on an individual basis.

During exercise-induced fatigue, between 70 and 80% of mechanical work is released as heat which the body stores; eventually, this will exceed the body’s heat loss capacity and result in a rise of core temperature [48]. During recovery after prolonged exercise, fitter individuals are able to maintain a warmer skin temperature due to thinner subcutaneous fat thickness and higher metabolic heat production [49], thus potentially benefiting more from cool vests.

One of the important outcomes of this study is the cool vest’s efficacy in improving post-exercise cooling, reducing skin and tympanic temperatures significantly below control levels. Given that greater cutaneous vasodilation occurs when Ts oscillates between 33 °C and 35 °C [50], it is likely that the cool vest induced vasoconstriction and a significant reduction in cutaneous blood flow and cooling of the tissues beneath. As the body temperature rises during exercise, increased skin blood flow facilitates heat transfer to the skin surface. This blood is subsequently cooled and returns via venous circulation, reducing core temperature [51]. Wearing the cool vest during different moments surrounding exercise, such as warm-up, stretching or during recovery, may reduce sweating rate, core and skin temperatures and may improve athletes’ perception of their own thermal state and skin wetness, producing a greater level of comfort [52] and potentially inducing beneficial effects on heart rate, VO_2_ and performance [53], although some of the outcomes could be affected by a placebo effect described by some psychological differences.

In our study, tympanic temperature increased during warm-up and exercise and tended to decrease after exercise finished. All this suggests that tympanic temperature shows an opposite evolution to skin temperature. The main reason is because the first will be more related to core temperature, while the latter will manifest the surface temperature, which is the result of thermoregulatory mechanisms activated to regulate core temperature as a compensatory mechanism. In this sense, the ability of the active musculature to generate heat, which requires a constant supply of blood to ensure activity, must be compensated by the ability to dissipate the temperature increase generated during its contraction. For this reason, distal parts of extremities have a relevant role in heat exchange due to the location of the arteriovenous anastomoses [43], which leads to heat loss triggered by transferring blood to the upper parts of the skin when exercising under hot conditions. Consequently, a boost in heat conduction provokes cooled blood into returning to the core. This results in a direct link between the use of the cooling waistcoat and a decrease in tympanic temperature, which means that it can be used as a reflection of the central temperature evolution. 

Therefore, we can affirm that the methodology (tympanic vs. skin) selected for monitoring body temperature will condition the results obtained, both during physical exercise and during recovery. In addition, it should be considered that the skin temperature calculated using the formula proposed by Ramadan [35] is a compound of the right scapula, left chest, upper left arm, right anterior thigh and left calf temperatures (see Figure 1). This means that the result obtained depends greatly on the movement pattern assessed and its symmetry, as each pattern will activate different muscle groups and a different amount of musculature in each of the body hemispheres [54]. Furthermore, multiplying the results obtained by a different correction factor (0.3 for the trunk and 0.2 for the legs) will more clearly reflect the changes experienced in the upper half of the body.

It is also suggested that lower oxygen consumption after halftime cooling is due to vasoconstriction of the peripheral vasculature because of an increased central blood volume, helping the oxygenation of muscles recruited in exercise and thus improving energy production due to the removal of blood lactate “which collaboratively benefit endurance performance by delaying the onset muscle soreness (DOMS)” [55]. One of the reasons why cooling at halftime may be a good option is the quantity of body fluid lost. Less fluid lost after cooling during recovery might have, as a benefit, a slower rate of dehydration during subsequent exercise, which may delay the presence of fatigue in muscular endurance and improve the capacity to maintain maximal aerobic power during exercise [13].

Despite these benefits, different studies employing cooling methods did not find differences in biomarkers such as blood plasma markers, lactate levels, creatine-kinase levels and blood plasma cytokines [56,57].

In relation to the type of methodology used for cooling, although other methods such as partial-body cryotherapy and cold-water immersion could achieve a faster decrease in skin body temperature [58], the use of the cool vest during the halftime of a football match has several advantages in relation to other methodologies, since it is a quick and easy method that allows players to keep their attention on their trainer’s comments during the halftime period. In addition, its use could be initiated on the way to the dressing room, which would allow optimization of its application time.

Several limitations must be considered in this study: (i) The absence of performance-related outcome measures (e.g., jump height, sprint performance, or functional markers such as MVC, RFD, etc.) raises the question of whether the cooling vest only affected skin temperature (without necessarily leading to improvements in performance). (ii) A crossover design could better account for individual differences. However, the parallel-group approach balanced scientific rigor with practical constraints in athlete recruitment and laboratory resources. Future studies with dedicated funding might implement crossover designs to extend our findings. (iii) The results obtained in this study are only applicable to the type and model of vest used in this study. Future studies should be conducted to confirm the effect of other makes and models. (iv) The temperature and relative humidity conditions in which this study was conducted are different from those which players are subjected to during competitions, so it is expected that certain differences, such as displacement intensities, will appear when this protocol is applied in real competition conditions. (v) Although tympanic temperature is a valid methodology for monitoring body temperature, there are more accurate methodologies that could be used. In our study, we decided to select these because of their low cost and ease of application, which makes it easier for other research groups to perform similar studies under different conditions. (vi) The heterogeneity in participants’ training backgrounds (i.e., sport type) may have introduced some variability in the investigated outcomes, which should be considered given that group allocation did not account for these differences. In any case, future studies evaluating the use of this methodology are needed. Although this study represents a preliminary assessment of the real efficacy of this methodology, we consider it necessary to evaluate its effects on neuromuscular responses, using tensiomyography, biomechanical analysis and the ability to repeat sprints, among other aspects. These evaluations should always consider individual muscle and mechanical responses.

## 5. Conclusions

The results of this study indicate that using a cool vest for 15 min after a simulated running football protocol on a treadmill changes the Ts pattern after minute 10 in a way that enhances thermoregulatory changes in skin temperature and maintains skin temperatures of the cool vest group at significantly lower values than the control group. Using a cool vest during recovery after exercise in mild temperatures seems to help the body to maintain a lower skin temperature, which might help performance in a posterior bout of exercise. Thus, it may be an appropriate intervention to assist halftime recovery and benefit on posterior performance. 

## Figures and Tables

**Figure 1 sports-13-00235-f001:**
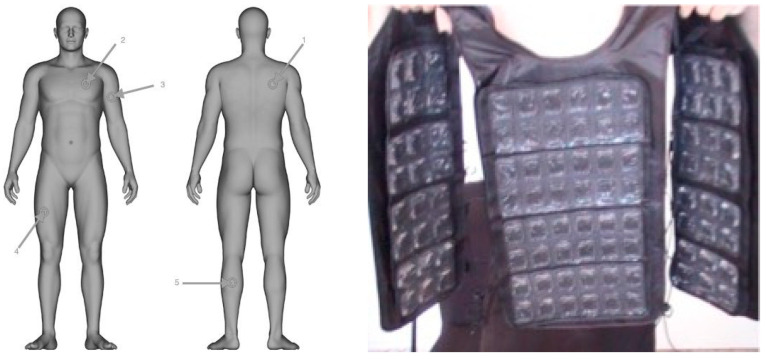
Details of the skin marker positions and the cool vest device. 1. Right scapula; 2. left chest; 3. left arm in upper location; 4. right anterior thigh; 5. left calf.

**Figure 2 sports-13-00235-f002:**
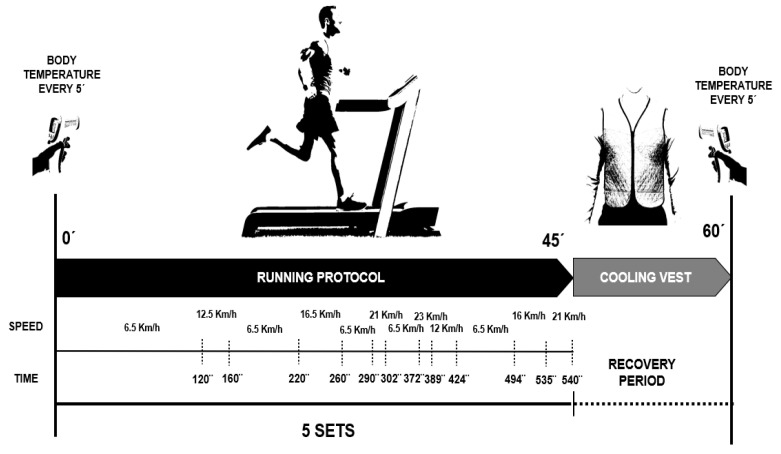
Exercise protocol: 45 min intermittent running session in a motorized treadmill.

**Figure 3 sports-13-00235-f003:**
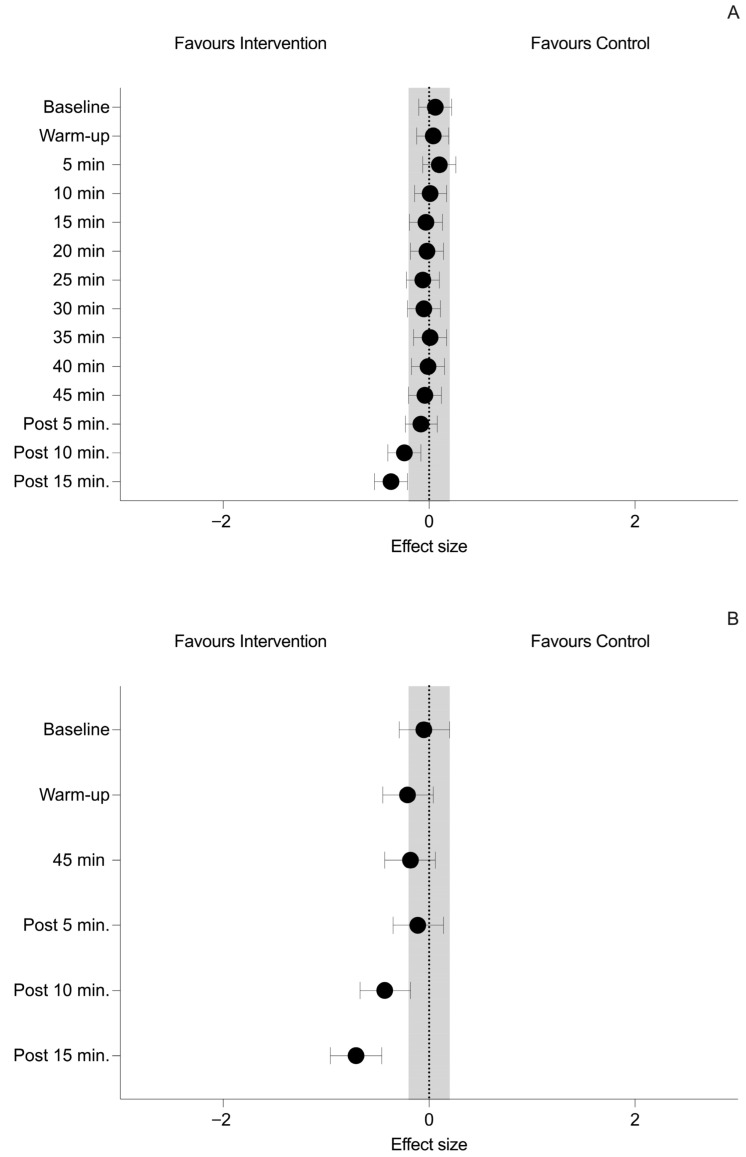
Between-group differences at all time points for (**A**) Skin temperature and (**B**) Tympanic temperature. Black dots indicate effect size and 95% confidence intervals. The grey area represents the upper and lower limits of a trivial effect size (i.e., −0.2 to 0.2).

**Table 1 sports-13-00235-t001:** Baseline characteristics of both groups.

Variable	IG (Mean ± SD)	CG (Mean ± SD)	*p*-Value	ES (95% CI)
Age (y)	22.46 ± 1.91	22.13 ± 1.66	0.534	−0.18 (−0.76, 0.39)
Height (cm)	178.38 ± 4.02	178.04 ± 4.90	0.799	−0.07 (−0.65, 0.50)
Weight (kg)	73.34 ± 4.48	73.82 ± 4.97	0.730	0.10 (−0.47, 0.67)
BMI (kg·m^−2^)	23.08 ± 1.67	23.30 ± 1.58	0.641	0.14 (−0.44, 0.71)
VO_2_max (mL·kg^−1^·min^−1^)	55.32 ± 3.39	55.97 ± 3.34	0.512	0.19 (−0.38, 0.77)

IG: Intervention group; CG: Control group; ES: Effect size; CI: Confidence intervals; y: years; BMI: Body mass index; VO_2_max: Maximal oxygen uptake.

**Table 2 sports-13-00235-t002:** Summary of mixed models results.

Time	Ts (°C) (Mean ± SD)		Tt (°C) (Mean ± SD)	
	Intervention group	Control group	Intervention group	Control group
Baseline	32.50 ± 1.15	32.25 ± 1.08	36.45 ± 0.50	36.44 ± 0.47
Warm-Up	31.72 ± 1.31	31.58 ± 1.27	36.30 ± 0.56	36.46 ± 0.67
5 min	30.78 ± 1.21	30.37 ± 0.72	--	--
10 min	30.36 ± 1.02	30.30 ± 1.11	--	--
15 min	30.32 ± 1.08	30.45 ± 1.05	--	--
20 min	30.48 ± 0.86	30.56 ± 1.01	--	--
25 min	30.39 ± 1.07	30.63 ± 1.10	--	--
30 min	30.39 ± 1.15	30.59 ± 1.17	--	--
35 min	30.63 ± 1.03	30.59 ± 1.22	--	--
40 min	30.60 ± 1.15	30.63 ± 1.33	--	--
Post 0 min	30.71 ± 1.25	30.86 ± 1.20	36.58 ± 0.51	36.62 ± 0.53
Post 5 min	31.59 ± 0.78	31.90 ± 1.18	36.29 ± 0.54	36.58 ± 0.64
Post 10 min	31.46 ± 0.67 *	32.40 ± 1.04	35.99 ± 0.45 *	36.54 ± 0.66
Post 15 min	31.07 ± 0.67 *	32.51 ± 0.91	35.70 ± 0.42 *	36.65 ± 0.58

* *p* ≤ 0.050 statistically significant differences from Control group. Data are presented as mean ± SD. CI: confidence intervals; Ts: Temperature of the Skin; Tt: Tympanic temperature.

## Data Availability

The data presented in this study are available upon request from the corresponding author due to ethical restrictions related to the protection of sensitive information and privacy of underage participants, in accordance with institutional and data protection regulations.

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
