# Peer review of "Effects of a Cooling Vest on Core and Skin Temperatures Following a Soccer-Specific Exercise Protocol†"

_sports, 2025, doi:10.3390/sports13070235_

Round 1
Reviewer 1 Report
Comments and Suggestions for Authors
I thank the authors and the editorial board for allowing me to review this manuscript. This study addresses the use of a practical and user-friendly recovery strategy, as it describes the effect of cool vests on skin and core temperatures after a soccer-specific exercise simulation. Although the manuscript may be viable for publication, substantial adjustments need to be performed throughout the document. Specifically, the statistical analysis may need to be re-run before the results and discussions can be reviewed. General and specific comments for each section can be found below.
TITLE
The current title may be somewhat misleading, as it seems to imply that the intervention occurred during an actual halftime (with a subsequent second half after that). Furthermore, defining the utilized exercise protocol as a “simulated soccer match” seems like a bit of a stretch. Given that the protocol was performed on a treadmill, lasted 45 minutes and lacked some crucial match-related elements (as you correctly mention at Lines 158-159), it may be better to describe it as a “soccer-specific exercise protocol” (or something similar). To ensure consistent terminology, edits may need to be made throughout the rest of the manuscript too. In summary, you could change the title to something like: “Effects of a cooling vest on core and skin temperatures following a soccer-specific exercise protocol”
INTRODUCTION
L 72-74: Considering that the effects of various cryotherapy modalities on muscle temperature and metabolism are not entirely consolidated in humans (see Kwiecien & McHugh, 2021: https://pubmed.ncbi.nlm.nih.gov/33877402/), I suggest that you use softer language here: “[…] might decrease inflammation in the musculoskeletal system [18], which may potentially reduce symptoms […]”
L 80-81: What do you mean by “cardiovascular states”? Please specify.
L 82-83: You state that “It still has to be established if a cooling intervention induces physiological benefits in team sports”. This sentence sounds quite generic, as it does not specify what “physiological benefits” means. Furthermore, there are several articles that have investigated the use of different cryotherapy modalities in team sports. For example, just considering the last 2 years (2023-2025), a few these aspects have been assessed in football (https://pubmed.ncbi.nlm.nih.gov/37279300/ and https://pubmed.ncbi.nlm.nih.gov/39593644/), basketball (https://pubmed.ncbi.nlm.nih.gov/39179229/) and rugby (https://pubmed.ncbi.nlm.nih.gov/37091363/). Several other articles have been published in previous years. Therefore, I believe you should contextualize your findings by providing a general background of previous research (e.g., by acknowledging that previous research already exists).
METHODS
General comments
In the section named “Experimental approach to the problem”, the study design is not entirely clear. You state that “a simulated soccer match” was performed. However (as stated in the “Exercise Protocol” section), it appears that this was only 45 minutes (as I previously mentioned regarding the study title). Furthermore, it is not specified when the cool vest was worn, and for how long (was it the entire duration of the 15-minute period after exercise?). These aspects should also be specified later (in the “Intervention group” paragraph).
Regarding the Exercise Protocol (Lines 149-162), I strongly suggest including an image with a timeline representing the different phases of the 45-minute simulated protocol (including the duration and speed of each step).
In the statistical analysis section, the decision to use two separate mixed models (i.e., one for temperature changes during the exercise protocol, and another for between-group comparisons during the recovery period) is somewhat unclear and may affect the quality of your data. Unless there is a specific reason for separating the models, this choice appears unjustified. A single mixed model for each dependent variable (which includes time and group as fixed effects) would allow for a better assessment of both within-subject changes and between-group differences over time. Splitting the analysis could mask potential time*group interactions, reduce statistical power, and increase the risk of Type I error (due to multiple testing). Unless a strong justification is provided, statistical analysis should be re-run using an integrated model, and changes should be made accordingly in the results and discussions (if necessary).
Specific comments
L 94-96: Please rephrase here, according to my previous comments regarding the title (i.e., this was not really a 15-minute "break", but rather the end of exercise). Also, it should be “on core and skin temperature” and “on a treadmill”.
L 97: This should be “allocated to two groups”.
L 97-100: I am a bit surprised to see that you did not choose a crossover design for your study. That seemed very feasible in this context, and it would have substantially contributed to increasing the quality of your study. Please add this aspect to the study limitations at the end of the discussion.
L 98: Although VO2max is an extremely common parameter in physiology-related research, please describe its meaning (maximal oxygen uptake) when mentioning it for the first time.
L 123: I suggest changing to “Participants in the IG group […]
L 137-144: I suggest renaming this section as: “Assessment of maximal oxygen uptake (VO2max)”. Furthermore, Information is missing regarding the equipment used for the incremental test. What type of metabolic mask and gas analyzer were employed? Additionally, please specify the criteria which determined the end of the test (e.g., voluntary interruption by the subject, due to exhaustion, VO2 plateau, threshold value for respiratory exchange ratio, or other).
L 142-144: It seems like you considered VO2max as the only variable according to which you assigned participants to either IG or CG. However, several other variables can influence the effectiveness of your intervention. For example, the following factors may introduce bias (if statistically significant differences are detected between groups): height, body mass and BMI (which can influence thermoregulation), age, regular physical activity levels (i.e., you describe your participants as “physically active males”, which may indicate participants with heterogeneous training backgrounds). Please consider adding a table in the “Participants” section, to compare baseline characteristics (age, height, weight, BMI and VO2max) between IG and CG. For each variable, report descriptive statistics (e.g., mean and SD) and between-group comparisons (e.g., p-values, effect sizes, and corresponding interpretation). In addition, please provide more details on the participants' training backgrounds. Specifically, you could indicate the type of sport or physical activity they were regularly engaged in prior to the study. For example: “The sample consisted of football players (n = ?), basketball players (n = ?), strength-trained individuals (n = ?), cyclists (n = ?), runners (n = ?), […]” This sentence should be added somewhere around Lines 108-109.
L 152-155: Aside from referencing prior research that analyzed time spent in different speed zones in soccer (reference 27), has this specific protocol been previously used in published studies, or did you specifically develop it for the present study? If previously used, please include appropriate citations to support its use.
L 165: Are you referring to °C (degree Celsius) here? If so, please report the unit of measurement correctly.
L 171-172: I suggest moving this last sentence to Line 168, for better logical flow. The update version would become: “The skin temperature was recorded from the right scapula, left chest, upper left arm, right anterior thigh and left calf (Figure 1). Specifically, Ts was calculated based on Ramathan’s equation: 0.3 (Chest + Arm) + 0.2 (Thigh + Calf) [29]. Data collection was carried out […]”
L 181: It is unclear how you extracted a value of d = 2.65 to perform power analysis. I checked the study cited here (reference 30), and I could not find this effect size value. Please specify.
L 192-193: How were data transformed, exactly? Please clarify (briefly).
RESULTS, DISCUSSION AND CONCLUSIONS
These sections will be re-assessed once comments regarding the statistical analysis have been addressed.
Reviewer 2 Report
Comments and Suggestions for Authors
Thank you for the opportunity to appraise the manuscript titled: “Effects of Halftime recovery with a Cool Vest Jacket on core and skin body temperature in a simulated soccer match” for Sports. Using an independent groups design, the authors have aimed to explore the effect of a practically meaningful cooling intervention (by means of a vest) applied after 45 min of unidirectional intermittent running on a treadmill. While the authors should be commended for interrogating an applied research question with a significant sample size, there are some major issues and fundamental flaws that unless rectified, would preclude acceptance of the submission. My comments now follow (where necessary using the continuous line numbering in the original submission).
The abstract would benefit from addition of absolute data to provide context for the findings.
The rationale largely omits the fundamental aspect of thermoregulatory challenge being a key modulator of fatigue and performance decrement (in the short term) and possible deleterious effects from a health perspective (if prolonged exposure). This is a major omission which needs remedy to fully contextualise the upcoming data. Otherwise, the reader is unclear on what the rationale for the intervention is – what are the authors trying to reverse/attenuate? Likewise, the magnitude of the effects, both in terms of heat challenge experienced by soccer players, and the potential for cooling – particularly after only 45 min of exercise – represents another fundamental omission.
L53-57: Largely redundant information – could omit this without influencing the rationale, therefore questioning it’s efficacy.
While the methods make mention of the adherence to ethical principles, no specific mention of ethical approval for this project is mentioned. This is a major concern.
Was exercise undertaken in a thermoneutral or hot environment? The methods do not contain mention of this crucially important fact, other than the conditions associated with the maximal testing beforehand.
A major methodological issue relates to the study design being an independent groups design rather than repeated measures. Was this approach taken for a particular reason other than maybe participant availability? Likewise, were baseline measures proven as similar between trials for all or selected variables? And, did the statistical power calculations account for this approach?
L123-130: Did the intervention group wear normal training attire plus the cooling vest, or just the vest? Plus did they remain seated, same for the control group?
L130: Typo
L140: Check unit presentation
L148: Duplication of data presentation is not necessary here
L149-161: Was the treadmill running based on a previously validated protocol? What about the test-retest reliability? Did it take place in the same conditions as previously mentioned?
When specifically was the intervention applied? How long did it take to apply? As the time-points are from the end of exercise, absence of such information hinders interpretation as 10 min post exercise may only be 8 min of intervention if it took a couple of minutes to dismount the treadmill, apply the jacket etc.
L214: Mentions effect sizes are presented during only the recovery phase, but given the study design, it would be useful to know effect sizes for all time-points.
L219: The results seem to start with a focus on the post-exercise window. While I understand the reasoning, I find it interesting to note the effects of the warm-up on temperature responses – were protocols effective to “warm-up” participants? Likewise, what about the comparability of physiological responses during exercise itself? Again, due to the study design, the reader needs faith that the protocols elicited the same effects irrespective of intervention.
Table 1 would benefit from being reformatted so the intervention and control group data is presented side-by-side for the same variables rather than being presented on a group basis. Also, the legend suggests “adjustment for baseline score” yet no symbol a is present in the table and why would data need adjusting if the groups are matched?
Figures should be significantly larger to allow them to be fully impactful. If figure 2 is presenting the same information as table 1, omit duplication.
Figure 2A is titled as between group differences, yet data are in the order of 32.5 deg C in both groups. I suggest revision of the title as differences suggest a differential between groups, which 32.5 deg C is not. Likewise, if these data are absolute data, it suggests that the participants are warmest at baseline – is this correct?
Small and moderate differences became apparent at 10 and 15 min, post-exercise respectively, with little difference occurring at 5 min. This therefore suggests that in the case of an ice vest, the longer the exposure the better. However, akin to comments about figure 2, the data is merely returning to baseline in the control group while remaining depressed in the intervention – thus causing the difference. So technically, no cooling is occurring, just attenuation of a return to baseline.
Would it not have been possible to include at least some performance measures after the half-time break to see the readiness that the intervention may (or may not) have afforded to performance in the second half?
Given the significant changes that addressing such issues will cause, I stop my review here given the likely impact this may have on the discussion section. However, if the editor sees fit to suggest a decision of revisions, I offer my services again for review of a resubmission.
Round 2
Reviewer 1 Report
Comments and Suggestions for Authors
I would like to thank the authors for improving the quality of their manuscript, especially in the introduction and methods sections. However, some changes still need to be made, especially within the discussion. My comments can be found below.
ABSTRACT
L 35: As specified in the first round of revisions, please avoid using the term “halftime”.
L 40-46: Please report p values, in addition to effect sizes.
L 48: I suggest rephrasing “contrasting with” as “compared to”.
INTRODUCTION
L 79: Please replace “throughout” with “through”.
L 82-85: I suggest rephrasing this part as: “[…] such as football [19,20], basketball [21] and rugby [22] – with some reporting positive outcomes in counteracting performance losses in speed and shooting accuracy, as well as reductions in creatine kinase concentrations in both football and rugby, there is still limited knowledge regarding the effects of using a cooling vest during short recovery periods, such as half-time in football.”
L 101 and 104-105: As previously mentioned, “15-minute break” is not the correct terminology here, as no exercise was performed afterwards (therefore, it was not a “break”).
METHODS
L 124-127: Thank you for including information regarding the participants’ training background. Given the heterogeneity in sporting profiles (e.g., strength-trained, endurance-trained, team sport athletes), there is potential for variability in how individuals responded to the intervention. While this does not compromise the results, it is worth acknowledging this aspect in the limitations (especially considering that you used a parallel-group design, rather than crossover). You could add the following limitation at the end of the discussion: “The heterogeneity in participants’ training backgrounds (i.e., sport type) may have introduced some variability in the investigated outcomes, which should be considered given that group allocation did not account for these differences.”
Table 1: Thank you for adding information regarding age, height, BMI and VO2max in both groups. Please also add information about weight (you can insert this between height and BMI). Furthermore, I believe that the 95% CI for Age should be in brackets, as follows: -0.18 (-0.76, 0.39). Finally, at Line 141, please correct the spelling from “grou0” to “group”.
L 162: Please replace the dot with a comma: “[…] at 6 km/h, followed by […]”
L 167: Please correct “requires” to “required”.
L 184-185: “[…] soccer players traying to know running distances and speeds, depending on the position of play.” I think this sentence needs to be clarified, as it is currently unclear.
L 188: This should be “allocated to two groups”.
L 216: I suggest rephrasing as: “Ultimately, the sample size was considerably higher, with 24 participants […]”
RESULTS
General comments
Looking at Table 2, I am a bit surprised with your findings. I would expect skin and tympanic temperature to 1) increase from “baseline” to “warm-up”; 2) remain elevated throughout the exercise protocol; and 3) gradually decrease during post-exercise recovery (in both IG and CG). However, it seems like skin temperature was higher at baseline than after warm-up, dropped during exercise, and rose again once exercise finished, which seems a bit odd. Any ideas of why this may have been the case?
Specific comments
L 250-253: Regarding skin temperature, you state that “[…] simple effects revealed non-significant differences (p > 0.05) in all time points except at Post 10 min. (ES =-0.24 (-0.40, -0.08, small)) and Post 15 min (ES =-0.37 (-0.53, -0.21, small)), the temperature was lower at the IG group.” Although you mention statistical significance, you do not report p values for Post 10 min and Post 15 min. The same observation is also valid for tympanic temperature. Please add such information.
Table 2: On the left side of the table, the word “Time” should be presented in bold, on the first row (i.e., the same row as “Ts” and “Tt”).
DISCUSSION
General comments
As it currently stands, it seems like this section discusses generic aspects related to temperature regulation, rather than addressing the specific intervention used in your study (i.e., cooling vest), how it may have improved thermoregulation, and why this matters from a practical perspective (i.e., effects on performance, etc.). This is evident, for example, at Lines 270-272, 288-293, 307-319 (as well as other parts of the discussion). While mechanistic explanations are certainly valuable, please make sure that they are used to help interpret and contextualize your findings (rather than just being included in isolation, without reference to your results).
Specific comments
L 270-272: You state that “The lower temperature reported in the intervention group could be associated with sweat evaporation for heat dissipation during exercise […]”. Currently, I don’t see how this statement is related to the use of the specific intervention in your study (i.e., cooling jacket).
L 273–275: This sentence is somewhat unclear and may cause confusion, by mixing up distinct physiological mechanisms. While increased catecholamine release during high-intensity exercise contributes to systemic vasoconstriction (which helps maintain blood pressure and redirect blood flow to active muscles) this generally limits heat dissipation (rather than promoting it). Specifically, elevated sympathetic tone can reduce skin blood flow, impairing the body’s ability to lose heat, as noted in the paper you referenced (Chou & Coyle, 2023). Therefore, I suggest rephrasing this sentence to more accurately reflect the physiological mechanisms involved and the role of catecholamines in this context.
L 275-276: “Although in general, the tendency was to attenuate the return to baseline of the skin temperature”. It is not entirely clear what this means. Please rephrase for better clarity.
L 279-281: Please specify (briefly) what you mean by “individual acclimatization”. Additionally, I suggest using softer language: “Therefore, different people exposed to the same temperature may display different stress responses. This could be attributed to […]”
L 282: I suggest replacing “big effect” with “substantial effect”.
L 285-286: I suggest rephrasing: “These differences in body size, configuration and composition may partly explain individual responses associated with cold exposure.” Is the initial meaning of the sentence preserved?
L 294-295: “the capacity of the cool vest to reduce the cooling rate to baseline levels”. This is not entirely clear. Looking at your data, I would say that the cool vest improved cooling (rather than “reducing” the cooling rate). Furthermore, I think you should primarily mention that using the cool vest reduced skin and tympanic temperature compared to the control group (rather than mentioning “baseline”).
L 295: Maybe it would be better to start this sentence as follows: “Given that greater cutaneous […]”
L 298: For better readability, I suggest splitting into two sentences: “[…] of the tissues beneath. As body temperature increases during exercise, […]”
L 298-299: It is unclear what this part means: “the demand of skin blood flow would increase, which would be cooled”. Please rephrase to ensure clarity.
L 301: Please replace “prior to exercise” with “surrounding” exercise, since you are also mentioning the recovery phase (which is not “prior to”, but rather after exercise).
L 302-304: I suggest using softer language: “may reduce sweat rate, core and skin temperature, and may improve athletes’ perception of their own thermal state and skin wetness, producing a greater level of comfort [53], and potentially inducing beneficial effects on heart rate, VO2 and performance [54], although some of the outcomes could be affected by a placebo effect […]”
L 307-319: As mentioned above, it is unclear why you included this information, since you are not using it to explain your findings. Such information can be helpful, but it needs to be clear how it could be a contributing factor to the observed results in your study.
L 320-324: This whole sentence is unclear. Please rephrase to improve clarity.
L 326: I suggest rephrasing as “during subsequent exercise”
L 329-330: I suggest rephrasing: “different studies employing cooling methods did not find differences in biomarkers as [...]". Furthermore, "blood plasma marker" sounds a bit odd, and I'm not sure what you mean exactly here.
L 335: This should be “has several advantages”.
L 339: I would say that the main limitation of this study is the absence of performance-related outcome measures (e.g., jump height, sprint performance, or functional markers such as MVC, RFD, etc.). This raises the question of whether the cooling vest only affected skin temperature (without necessarily leading to improvements in performance). Therefore, I strongly suggest that you add this limitation as the first one. You can then move on to the other limitations, which you have already described appropriately.
CONCLUSIONS
L 359-360: Didn’t you use the vest for 15 minutes after the exercise protocol? Why are you reporting 10 minutes here?
Reviewer 2 Report
Comments and Suggestions for Authors
Most of my original comments have been addressed somewhat satisfactorily. I suggest proof-reading to ensure unit presentation and formatting is improved throughout.
Author Response
Dear Reviewer,
Thank you for your valuable feedback regarding unit presentation and formatting consistency. We particularly appreciate your suggestion for thorough proofreading, as precision in these details strengthens scientific rigor.
To address your comment comprehensively, we have conducted a full-manuscript audit of units, symbols, and formatting conventions, rectified all identified inconsistencies across text descriptions, tables and figures, and statistical reporting.
Thank you again for your meticulous review—we welcome further suggestions to enhance clarity.